# Anthocyanin-Rich Extract Mitigates the Contribution of the Pathobiont Genus *Haemophilus* in Mild-to-Moderate Ulcerative Colitis Patients

**DOI:** 10.3390/microorganisms12112376

**Published:** 2024-11-20

**Authors:** Yannik Zobrist, Michael Doulberis, Luc Biedermann, Gabriel E. Leventhal, Gerhard Rogler

**Affiliations:** 1University of Zurich, 8006 Zurich, Switzerland; yazo@hispeed.ch; 2Department of Gastroenterology and Hepatology, Department of Medicine, Zurich University Hospital, 8091 Zurich, Switzerland; doulberis@gmail.com (M.D.); luc.biedermann@usz.ch (L.B.); 3Gastroklinik, Private Gastroenterological Practice, 8810 Horgen, Switzerland; 4Division of Gastroenterology and Hepatology, Medical University Department, Kantonsspital Aarau, 5001 Aarau, Switzerland; 5PharmaBiome AG, 8952 Schlieren, Switzerland; g.leventhal@pharmabiome.com

**Keywords:** ulcerative colitis, calprotectin, anthocyanin, microbiome, IBD

## Abstract

Anthocyanins (ACs) have been shown to elicit anti-inflammatory and antioxidant effects in animal models of ulcerative colitis (UC). Furthermore, we previously observed in a double-blind randomized trial in UC patients that biochemical disease activity tended to be lower in patients that were exposed to AC. Here, we report on the changes in the fecal microbiome composition in these patients upon AC exposure. UC patients received a 3 g daily dose of an AC-rich bilberry extract (ACRE) for eight weeks. We determined the microbiome composition in longitudinal stool samples from 24 patients and quantified the degree of change over time. We also correlated the relative abundances of individual microbial taxa at different timepoints to fecal concentrations of calprotectin, a proxy for inflammation. Microbiome composition did not change over time as a result of the intervention, in terms of both alpha and beta diversity. However, before the intervention, the abundance of *Haemophilus parainfluenzae* was positively correlated with fecal calprotectin concentrations, and this correlation persisted in placebo-treated subjects throughout the study. In contrast, the correlation between *H. parainfluenzae* and calprotectin vanished in ACRE-treated subjects, while the relative abundance of *H. parainfluenzae* did not change. Our results suggest that ACRE treatment mitigates the contribution of *H. parainfluenzae* to inflammation. Further research is warranted to better comprehend the role of microbial composition in response to medical therapy including AC-rich extract in UC patients.

## 1. Introduction

The incidence and prevalence of inflammatory bowel disease (IBD) and in particular ulcerative colitis (UC) is rising worldwide [1]. The exact pathophysiology of UC remains unknown, and various factors are thought to influence the emergence of UC [2], complicating early detection in cases of suspected IBD. The current non-invasive diagnostic gold standard is determination of fecal calprotectin. The latter represents a crucial non-degradable leukocyte protein with the best correlation to endoscopic inflammatory indices compared to C-reactive protein (CRP). It effectively distinguishes mild, moderate, and severe inflammation [3].

Approximately two-thirds of UC patients with mild-to-moderate disease activity can be successfully treated with the anti-inflammatory drug mesalamine (5-ASA). However, non-responders to 5-ASA treatment remain a clinical challenge, with severe cases requiring invasive treatments such as colectomies. Novel biologics are promising effective medications for IBD. Ustekinumab, a human anti-IL12/23p40 monoclonal antibody, has been shown to provide optimal rates of both deep mucosal healing and transmural healing, even in hard-to-treat patients (i.e., prior to colectomy) [4]. Nevertheless, biologics are far from ideal, with substantial short- and long-term toxicity risks and with high annual costs for newer treatment options [5].

Because of this, there is strong interest from IBD patients in safer complementary therapeutic options that are perceived as “natural and holistic” with fewer side effects [6]. For instance, lion’s mane (*Hericium erinaceus*) is an edible fungus that is known for its anti-inflammatory and antineoplastic properties in colorectal tissue. Given the complex biochemical nature of fungi, how these therapeutic properties are biochemically mediated remains unknown. Proposed mechanisms include immune system regulation or modification of the gut microbiota to increase short-chain fatty acid production [7,8].

Other natural products are more well-defined in the biochemical sense. Anthocyanins (ACs) are a specific type of deglycosylated anthocyanidins that are concentrated in various vegetables and berries, particularly bilberries [9,10]. ACs are known to have antioxidant and anti-inflammatory effects [10,11] and have thus been preclinically investigated for the treatment of UC. Animal model studies have shown that AC treatment has beneficial effects on dextran sodium sulfate (DSS)-induced colitis. These effects include lower histological scores, reduced cytokine release, less colonic shortening (fibrosis), less weight loss, less hepatosplenomegaly, increased intestinal permeability (which favors bacterial translocation), and lower abundances of pathogenic bacteria compared to control groups [12,13].

Mechanistically, it is feasible that ACs can act on the microbiome in the colon. The majority of ingested ACs bypass absorption in the stomach and small intestine and reach the microbiome-rich colon. There, certain bacteria deglycosylase and metabolize ACs into phenolic acids such as protocatechuic, vanillic, syringic, gallic, or 4-hydroxybenzoic acid [14]. These bacterial fermentation derivates exhibit beneficial chemoprotective effects due to their intrinsic anti-inflammatory and antioxidative properties [15]. Interestingly, positive effects of AC such as increased short chain fatty acids (SCFAs), reduced spleen weight, re-extension of colon length, or ameliorated histological scores could not be achieved or replicated in germ-free mice [12], suggesting a key role of the microbiome in the AC mode of action. Intake of AC also alters the intestinal microbial composition [13,16].

Another mechanism by which ACs act as antioxidant and anti-inflammatory agents on IBD is via secretory immunoglobulin A (IgA). IgA is produced by antibody secretory cells locally in the gut lumen and forms dimers. On the one hand, IgA possesses a crucial pivotal role in shaping the gut microbiome composition and maintaining homeostasis within the intestinal immune system. On the other hand, derangements in IgA production, secretion, and/or function may occur during pathological conditions such as IBD, the pathogenesis of which remains largely unknown [17]. In a relevant clinical study, oral ACRE administration was beneficial for the management of oxidative stress and inflammation, an effect that was attributed to an increase in IgA, antimicrobial beta-defensin 2, as well as anti-inflammatory IL-10 [18].

We recently posted a preprint (currently in peer review) that reports on a multicenter, double-blind, randomized, placebo-controlled phase IIa study [19] to confirm the therapeutic effect of “AC-rich extract” (ACRE) therapy previously observed in a pilot study [20]. In this study, we observed that fecal calprotectin concentrations decreased during treatment with ACRE and subsequently increased again after stopping ACRE therapy [19]. Here, we asked whether the effect of ACRE on fecal calprotectin was mediated by the microbiome. We hypothesized that the effect acted on the microbiome either directly, by inhibiting or promoting certain bacteria, or indirectly, by modulating potential negative effects of certain microbiota. To answer this, we characterized the bacterial microbiome composition in the UC patients throughout the study and particularly investigated whether specific genera might be associated with the observed effect of fecal calprotectin reduction during the intervention.

## 2. Methods

### 2.1. Ethics

The study was carried out in accordance with principles enunciated in the current version of the Declaration of Helsinki [21], the guidelines of Good Clinical Practice [22] issued by International Council for Harmonization of Technical Requirements for Pharmaceuticals for Human Use, and the Swiss regulatory authority’s requirements. The project was approved by the Ethics Committee Zurich (BASEC-Nr, 2017-00156). Written informed consent was obtained by all patients before randomization.

### 2.2. Study Design

The full protocol and study design can be found in the Appendix A. Briefly, we included individuals over 18 years of age who had a UC diagnosis for at least three months with a modified Mayo score of 6–12 and disease activity despite therapy with 5-ASA and steroids. A detailed description of all inclusion and exclusion criteria can be found in the Appendix A. Of note, the modified Mayo score was initially introduced by the Food and Drug Administration (FDA) as “Guidance for Clinical Trial Endpoints”. Specifically, it refers to the “endoscopy subscore” of the Mayo score, which should be modified so that a value of 1 does not include friability. This is due to the fact that existence of friability (even if graded as mild by the endoscopist/central reader) is not consistent with clinical remission [23,24]. A Mayo score of 5 or below indicates mild disease activity, a score between 6 and 10 signifies moderate activity, and a score from 11 to 12 represents severe disease activity [25].

Study participants were randomly assigned to either the verum or placebo group in a 2:1 ratio. The verum group received a three times daily dosage of 2 × 500 mg of an ACRE provided by Walther Riemer GmbH (Nimbo Green, Ningbo, China), corresponding to 100 g dried bilberries or 840 mg of anthocyanins per day. Regarding the composition of the administered ACRE, it was an ethanoic bilberry extract with most prominent ingredient being cyanidine-3-O-glucoside. The remaining active ingredients were cyanidine-3-O-galactoside, cyanidine-3-O-arabinoside, delphidin-3-O-galactoside, delphidin-3-glucoside, delphidin-3-arabinoside, petunidin-3-galactoside, petunidin-3-glucoside, petunidin-3-arabinoside, peonidin-3-O-glucoside, peonidin-3-O-galactoside, peonidin-3-O-arabinoside, malvidin-3-galactoside, malvidin-3-glucoside, and malvidin-3-arabinoside.

The screening phase spanned four weeks, followed by an eight-week intervention period and a subsequent three-week follow-up phase. Three consultations were conducted during the intervention [19], with stool samples and other relevant data collected at all timepoints. A detailed study procedure description is available in the Appendix A.

### 2.3. Preanalyticss

The stool was collected with an OMNIgene^®^ GUT kit (DNA Genotek Inc., Ottawa, ON, Canada) and later stored at −20 °C until DNA extraction and sequencing (performed by Microsynth AG, Balgach, Switzerland). Amplicon sequencing was performed using Illumina MiSeq paired-ends sequencing technology. The hypervariable region V4 of bacterial 16S rRNA genes was amplified using the primers 515F (GTGCCAGCMGCCGCGGTAA) and 806R (GGACTACHVGGGTWTCTAAT) to generate an approximate amplicon size of 300 bp. Multiplexed libraries were constructed with help of the two-step PCR protocol (as recommended by Illumina, San Diego, CA, USA) [26]. The paired-end reads were trimmed of their locus-specific primers and subsequently merged to in silico reconstruct the amplified region, resulting in a total of 22.74 million reads, from which amplicon sequence variants (ASVs) were then identified.

ASVs were inferred using Dada2 v1.18.0 with read length filtering set to 250 and 210 and maximum expected errors (maxEEs) set to 4 and 5 for forward and reverse reads, respectively, and inference performed in “pseudo pool” mode. Read pairs were merged with minimum overlap (minOverlap) of 20, and bimeras were removed using the consensus method. The prepared GTDB r95 taxonomic database for Dada2 (GTDB ref, Dataset Ref) was used for taxonomic annotations via the assign taxonomy function in Dada2.

### 2.4. Statistics

We obtained ASVs from 35 subjects at different study timepoints. After excluding data from individuals lacking baseline calprotectin measurements or with data available at fewer than 3 study timepoints, our final cohort comprised 24 patients, with 17 in the verum group and 7 in the placebo group.

Genus-level relative abundances were computed by summing the counts of all ASVs that were assigned to a specific genus and normalizing by all reads. Whenever log-transformed relative abundances were computed, a pseudo count of 1 was added to the read counts.

The Shannon diversity, *H*, was computed at the relevant level of taxonomic resolution (e.g., genus). Subsequently, resulting effective number of genera was computed as the exponential of the Shannon diversity, *exp(H)*, and represents the equivalent number of evenly distributed taxa with the same Shannon diversity. Comparisons between treatment groups were performed for each timepoint individually using Wilcoxon signed-rank tests.

We computed the Aitchison distance, i.e., the magnitude of change in microbiome composition between the center-log-ratio-transformed relative abundances [27]. Comparisons between groups or timepoints were performed using Wilcoxon signed-rank tests.

The change in individual taxa (e.g., genera) over time following the baseline was estimated from a linear mixed model of the log_10_-transformed relative abundances with treatment as a fixed effect and by treating timepoint as an integer including a random slope for the effect of time and a random intercept for each individual.

To quantify the association of individual microbiome genera with calprotectin levels prior to the intervention, we performed individual linear mixed model regressions of the log_10_-transformed calprotectin concentrations on the log_10_-transformed relative abundances for those genera that were detected in at least 50% of the samples. To capture potential variability in the microbiome over time, we used both the screening and baseline abundances and accounted for subject identity as a random effect. We adjusted the *p*-values using the method of Benjamini–Hochberg [28].

## 3. Results

We previously reported [19] that ACRE treatment reduced fecal calprotectin concentrations resulting in a significantly lower concentration in the ACRE group compared to the placebo group by the third visit but not at earlier visits. In order to better understand dynamics of the fecal calprotectin over time, we estimated a per-subject change in fecal calprotectin using a linear mixed model. Fecal calprotectin decreased steadily during treatment in the ACRE group (*p* = 0.0243) but did not change in the placebo-treated patients (*p* = 0.427; Appendix A). The concentration subsequently increased again following the end of the treatment. This suggests that ACRE treatment might act gradually and consistently.

A gradual action of ACRE on the microbiome might manifest in different ways. First, in a direct way, ACRE treatment might gradually shift the microbiome composition over time away from a proinflammatory state. Second, in an indirect way, ACRE treatment might not affect the microbiome composition per se, but instead mitigate any proinflammatory impact of the microbiome. In the first case, we would expect the microbiome composition to gradually change during ACRE treatment—but not during placebo treatment—and in the second case we would expect the microbiome to not change in either group. We did not observe any evidence of a systematic change in microbiome composition in either treatment group. We quantified the change in microbiome composition in terms of alpha and beta diversity. For alpha diversity, we modeled the change in the effective number of genera using the same linear mixed model approach as for fecal calprotectin with a treatment-specific slope and common intercept at baseline and per-subject random slopes and intercepts. Alpha diversity did not change over time in either treatment group (Figure 1a). For beta diversity, we quantified the degree of change in terms of the Aitchison distance between baseline and Visit3 for each subject and compared this to the distribution of distances between subjects at baseline. The Aitchison distance between baseline and Visit3 was not significantly different in the ACRE and placebo groups (*p* = 0.697, Figure 1c), implying that individual patients mostly retained their microbiome composition during treatment (Figure 1b). Taken together, we do not find evidence that ACRE treatment modifies the microbiome composition directly. The same assumption can be drawn also for the antioxidant and anti-inflammatory impact of ACRE on fecal calprotectin. ACRE might pleiotropically affect the inflammation by downregulating proinflammatory cytokines such as interleukins (IL)-1b and IL-6 and tumor necrosis factor (TNF) and upregulation of anti-inflammatory interleukin 10 [29].

Given that ACRE treatment does not modify microbial composition directly, we next turn to the second option where ACRE acts indirectly, implying that there are certain microbiota characteristics that promote inflammation that are then mitigated by ACRE treatment. To identify such characteristics, we analyzed whether certain genera correlated with fecal calprotectin concentrations before the start of the intervention. To increase statistical power, we grouped the screening and baseline samples together but accounted for subject identity as a random effect.

We identified 13 genera whose relative abundances were either positively or negatively associated with calprotectin levels prior to the intervention (Figure 2a). Of these, only two—*Haemophilus* and *Parasutterella*—remained significant after correction for multiple testing at a false discovery rate below 0.1 (Appendix A). Increasing relative abundances of the genus *Haemophilus* were associated with higher concentrations of fecal calprotectin (*p*_adj_ = 0.053, conditional R^2^ = 0.43), whereas higher relative abundances of *Parasutterella* were associated with lower concentrations of calprotectin (*p*_adj_ = 0.053, conditional R^2^ = 0.52) (Figure 2b).

This analysis identifies *Haemophilus* abundance as a potential contributor to inflammation and conversely *Parasutterella* as a potential mitigator of inflammation in the subjects before initiating treatment. We presumed that if these contributions were robust, then they should persist throughout the study in the placebo-treated individuals in which we do not expect any modulation of the interaction between microbiome and inflammation.

The association of *Haemophilus* and fecal calprotectin persisted in the placebo group throughout the intervention, while the association for *Parasutterella* did not. We used all sampling points after the start of the intervention, i.e., Visit1–3 and follow-up, and accounted for subject identity as a random intercept. *Haemophilus* relative abundance remained significantly correlated with fecal calprotectin concentrations in the placebo group after the start of the intervention (*p* = 0.0347, conditional R^2^ = 0.367, Appendix A). In contrast, the negative correlation between *Parasutterella* relative abundance and fecal calprotectin was not significant after the start of the intervention (*p* = 0.673, conditional R^2^ = 0.28). This suggests that *Haemophilus* might indeed contribute directly to inflammation.

We hypothesized that if ACRE directly exerts an influence on the microbiota, for instance, by depleting *Haemophilus* abundance, then a correlation between *Haemophilus* and fecal calprotectin would be expected to remain in the ACRE group, with a potential overall decrease in *Haemophilus* abundance. Alternatively, if ACRE were to modify the mechanism with which *Haemophilus* contributes to inflammation, then we would not expect an effect on *Haemophilus* abundance but rather that there would nevertheless be a decrease in fecal calprotectin concentration.

To test these hypotheses, we first investigated whether the relative abundance of *Haemophilus* changed during ACRE treatment. Second, we investigated whether the relationship between the abundance of *Haemophilus* and fecal calprotectin concentration remained in the ACRE group after the start of the intervention.

ACRE treatment did not impact *Haemophilus* abundance but did negate the association between *Haemophilus* and fecal calprotectin. The estimated change in *Haemophilus* abundance over time in the ACRE group was not significantly different from zero (*p* = 0.345, Figure 3). However, fecal calprotectin was no longer significantly associated with *Haemophilus* relative abundance after the start of the intervention (*p* = 0.422, conditional R^2^ = 0.42) (Figure 4). Closer inspection of the ASVs that were identified as *Haemophilus* in our data revealed only a single ASV that mapped with 100% identity to the strain *Haemophilus parainfluenzae* ATCC 33392. Taken together, these results indicate that ACRE treatment modifies the interaction between *Haemophilus parainfluenzae* and inflammation, rather than by directly decreasing *Haemophilus* abundance.

## 4. Discussion

In our study we aimed to investigate to what degree the previously observed effect of ACRE treatment on lowering fecal calprotectin concentrations in a clinical study in subjects with mild to moderate UC was linked to a modulation of the microbiome in these subjects [19]. Unexpectedly, we did not observe any substantial shifts in microbiome composition induced by ACRE treatment. Instead, we observed that ACRE treatment mitigated the association between the abundance of the genus *Haemophilus* in the fecal microbiome and the fecal calprotectin concentration. Prior to initiating the study, patients with higher abundances of *Haemophilus* also had higher concentrations of fecal calprotectin and this correlation persisted in the placebo-treated patients. In contrast, the correlation vanished during ACRE treatment. Therefore, ACRE administration did not reduce fecal calprotectin concentrations by directly modulating the microbial composition, but rather by indirectly affecting the proinflammatory mechanisms of bacterial taxa such as *Haemophilus*.

In line with our findings, we observed no difference in systemic inflammation, such as CRP levels, between treatment arms in our previous trial [19]. Interestingly, although ACRE appears to act locally rather than systemically, other studies have shown that inflammation-modulating interventions, like dried bilberries in a colitis model, can reduce both local and systemic inflammatory markers, such as TNF and interferon-γ. Furthermore, the proinflammatory activity of *Haemophilus* across the gut–lung axis emphasizes the potential of localized treatment impacts on inflammation without directly altering microbial composition [30].

Pathogenic bacteria of the upper respiratory tract like *Haemophilus* can interact with the gut microbiota along the so-called gut–lung axis. After birth, both the lungs and the gut are exposed to orally ingested microorganisms that contribute to the shaping of a stable and complex equilibrium between lung and gut flora. Gut microbes influence immune responses locally, systemically, and in the lungs, affecting conditions like asthma, allergic responses, and chronic obstructive pulmonary disease [31]. Of note, in a relevant recent study with lung adenocarcinoma patients, *Haemophilus parainfluenzae* was the most commonly found species shared between the lung and gut microbiota [32].

The observation that treatment with AC did not affect the microbiome composition is unexpected based on previous mouse studies in the literature showing AC impacting microbiome alpha and beta diversity in UC. Two studies observed an increased Shannon index and a significant beta diversity shift after ingesting 200 mg/kg AC for 7 or 17 days, respectively [13,33]. Another four-week study with daily intake of 3.47mg suggested a potential increase in alpha diversity post anthocyanin intake [34]. However, these studies were performed in mice with DSS-induced colitis, used sources of AC other than bilberries that may impact the effect [35], and had higher daily doses per kg body weight (173–200 mg/kg) compared to our study (ca. 11.8 mg/kg; assuming an average European weight of 71 kg).

The genus *Haemophilus* has previously been reported to be increased in abundance in IBD patients [36,37,38,39]. The positive correlation we observed between the relative abundance of *Haemophilus* and fecal calprotectin concentrations is in line with a previous work that found an association of *Haemophilus* with more severe disease [40].

Consistent with our results, two studies have reported an increased presence of *Haemophilus* spp. in UC patients that is particularly pronounced in active stages [41,42]. Thus, emerging evidence suggests that *Haemophilus* spp. may be considered as a potentially pathogenic genus for UC. Moreover, studies have demonstrated that *Haemophilus parainfluenza* exhibits strong IgA coating in individuals with UC [43]. IgA, the predominant dimeric antibody in the intestinal mucosa, plays a crucial role in coating and neutralizing pathogens. IgA coating is also observed in endogenous microbiota, though to a lesser extent than with pathogens [44]. Given that certain genera influencing UC severity share similarities with pathogens [45], Palm et al. and Shapiro et al. suggest pathogenic microbiota may be more heavily coated with IgA [46,47]. Furthermore, more active UC is associated with increased IgA coating and higher fecal IgA levels [48] and, on the other hand, mice with elevated IgA levels and therefore more coated bacteria demonstrate greater resistance to DSS-induced murine colitis [49]. Finally, blueberry ingestion is linked to increased IgA secretion [50,51].

In summary, it is reasonable to hypothesize that IgA secretion is increased by higher inflammatory activity or pathogenic microbes, thus coating the more pathogenically active microbiota, trying to provide control of colitis. ACs, in turn, seem to facilitate IgA production, exerting their potentially anti-inflammatory effect on UC. ACRE may stimulate IgA secretion, resulting in a more intensive coating of pathobiont *Haemophilus*, thereby reducing its proinflammatory potential. However, other potential anti-inflammatory mechanisms of ACRE via the microbiome are conceivable, such as promoting beneficial SCFA production, reducing colonic shortening, or enhancing epithelial barriers [12,16].

Regarding the interpretability and future implications of the present work; the observation of a clinically meaningful decrease in a key biochemical parameter that is linked to disease activity in UC suggests that AC might be considered as a viable treatment option in UC that warrants further evaluation. While it is unlikely that anthocyanins exert their effects exclusively by mitigating the proinflammatory potential of *Haemophilus* spp., their potential benefits may be more pronounced in individuals with higher *Haemophilus* spp. abundance in their intestinal microbiome. Additionally, probiotic therapies might prove valuable in reducing pathobionts such as *Haemophilus* spp. in UC patients, fostering a more favorable microbial microenvironment for the disease. Nevertheless, larger studies are essential to accumulate positive evidence and secondarily reach a consensus on the composition of the microbiota and its analysis in order to take a step towards personalized medicine, especially with complementary options such as ACRE.

A particular strength of our study is the randomized double-blind design with a novel intervention and it is among the first to investigate the impact of a high-dose anthocyanin intervention on the microbiome composition in UC patients. While there are already some investigations on this topic in animal models [12,13,16,33,34,35], this is to the best of our knowledge the first human study. Another strength of our study is that we repeated assessments of the key parameters, microbiome composition and fecal calprotectin concentration, over time for the same subjects that enables us to account for some of the intra-individual variability.

## 5. Limitations

Several limitations of our study have to be acknowledged. Firstly, our sample size comprising only 24 patients is rather small and for this reason we have taken care that our statistical inferences are appropriate. A further limitation that is shared by most other microbiome studies is that our work considers only relative abundances of microbial taxa and focuses primarily on fecal calprotectin. We did not investigate other related parameters such as the microbial metabolome or immunologic factors like intestinal IgA production. Finally, we exclusively focused on bacterial taxa of the intestinal microbiota and did not take other microbes into account, such as fungi, viruses, or protozoa.

## 6. Conclusions

In conclusion, we provide evidence that the anti-inflammatory effect of the ACRE intervention was not mediated via a relevant modulation of the microbiome, as originally thought, but rather via a regulation of the proinflammatory effect of its pathobionts such as *Haemophilus parainfluenzae.* Whether this effect is directly caused by the administered ACs or rather by their degradation products after metabolization by the microbiome remains unclear. Given IgA’s dual role in maintaining gut homeostasis and its involvement in inflammatory processes in IBD, it is plausible that ACs exert anti-inflammatory effects through the modulation of IgA secretion. This action could support intestinal immune balance, potentially attenuating inflammation. Nonetheless, the specific interactions between AC intake, IgA dynamics, and the microbiome in UC are not yet fully elucidated, underscoring the need for further research to clarify these mechanisms.

## Figures and Tables

**Figure 1 microorganisms-12-02376-f001:**
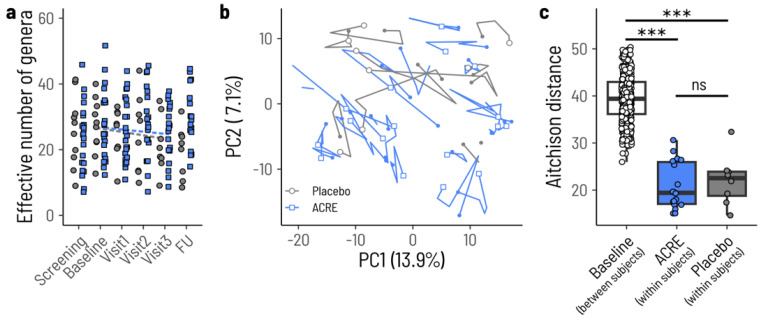
Microbiota composition is not affected by ACRE treatment. (**a**) Microbiome alpha diversity does not change during treatment. Blue squares and grey circles show the alpha diversities at different timepoints in the study for the ACRE and placebo groups, respectively. Symbols from the same patients are connected by lines. The dashed lines show the estimated slopes from a linear mixed effects model that are not significantly different from zero (*p* = 0.260 and *p* = 0.215). (**b**) Principal component analysis (PCA) of the fecal microbiota based on genus composition over time. Each path represents an individual patient (baseline: filled shapes, follow-up: open shapes). (**c**) Aitchison distances between subjects at baseline, or between baseline and Visit3 for the bilberry and placebo groups. ns = non-significant, *** = *p* value < 0.00005.

**Figure 2 microorganisms-12-02376-f002:**
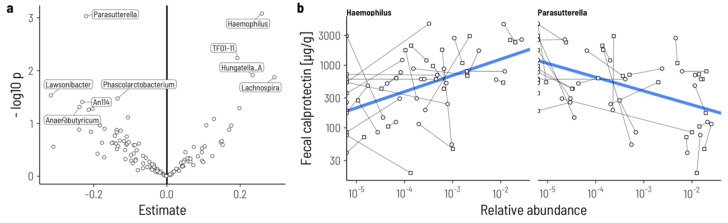
Genera associated with high/low fecal calprotectin concentration prior to the intervention. (**a**) Volcano plot of the regression slopes and their individual *p*-values. Genera with *p* < 0.05 are labeled. (**b**) Visualization of the relationship between relative abundance (*x*-axis) and fecal calprotectin concentrations for the two genera with a false discovery rate < 0.1. Samples from the same individual at screening (circles) and baseline (squares) are connected with a line.

**Figure 3 microorganisms-12-02376-f003:**
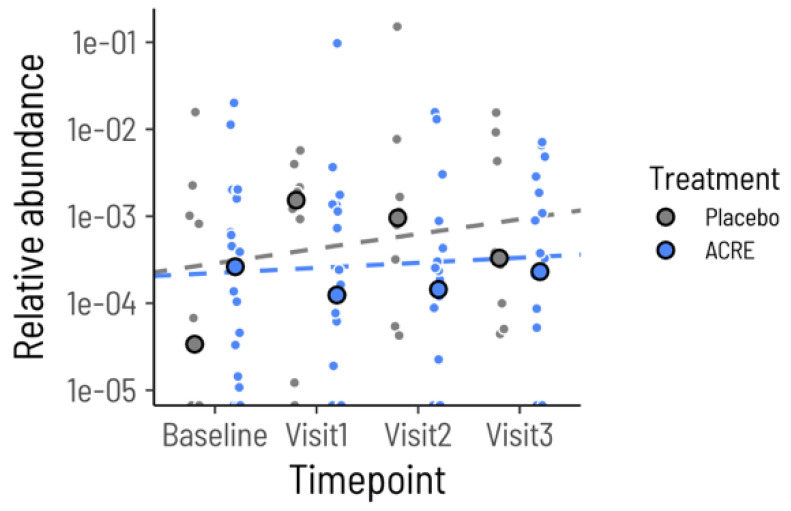
*Haemophilus* relative abundance remains stable during treatment. Each small circle shows the relative abundance of the genus *Haemophilus* in either the placebo group (grey) or ACRE group (blue). The large circles show the mean within the groups for each timepoint. The dashed lines show the estimated regression slope of log_10_ relative abundance over time.

**Figure 4 microorganisms-12-02376-f004:**
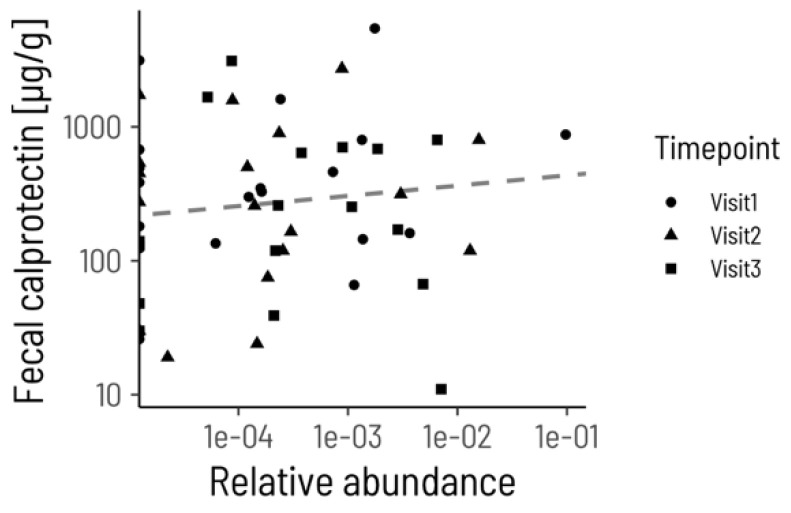
*Haemophilus* relative abundance is not significantly correlated with fecal calprotectin during ACRE treatment. Each point corresponds to a sample from a patient at Visit1 (circles), Visit2 (triangles), or Visit3 (squares). The dashed line shows the estimated regression slope from a linear mixed model with subject as a random effect.

## Data Availability

The original contributions presented in the study are included in the article/Appendix A, further inquiries can be directed to the corresponding author.

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
