# Peer review of "Anthocyanin-Rich Extract Mitigates the Contribution of the Pathobiont Genus Haemophilus in Mild-to-Moderate Ulcerative Colitis Patients"

_microorganisms, 2024, doi:10.3390/microorganisms12112376_

Round 1
Reviewer 1 Report
Comments and Suggestions for Authors
Interesting study, nevertheless several aspects are suggested to be completed.
1. Please provide the composition of ACRE. Is ACRE standardized?
2. The authors have adequately indicated the limitations of their study. As the paper would require further in-depth research in this area on a larger group of patients, a change in article type to short communication is suggested.
Author Response
Reviewer 1
Interesting study, nevertheless several aspects are suggested to be completed.
Response: We thank Reviewer 1 for his/her enthusiasm as well as the accompanying constructive comments
- Please provide the composition of ACRE. Is ACRE standardized?
Response: Thank you for your valuable comment. In the revised version of the manuscript, we provide the following further piece of information about composition of ACRE:
Regarding the composition of the administered ACRE; it was an ethanoic bilberry extract with most prominent ingredient Cyanidine-3-O-glucoside. Rest of the contained active ingredients were Cyanidine-3-O-galactoside, Cyanidine-3-O-arabinoside, Delphidin-3-O-galactoside, Delphidin-3—glucoside, Delphidin-3-arabinoside, Petunidin-3-galactoside, Petunidin-3-glucoside, Petunidin-3-arabinoside, Peonidin-3-O-glucoside, Peonidin-3-O-galactoside, Peonidin-3-O-arabinoside, Malvidin-3—galactoside, Malvidin-3—glucoside and Malvidin-3—arabinoside.
The above-mentioned information has been added now to the revised Version of the manuscript, according to the reviewer´s 1 suggestion.
Page 4, line 131-137
- The authors have adequately indicated the limitations of their study. As the paper would require further in-depth research in this area on a larger group of patients, a change in article type to short communication is suggested.
Response: We totally agree with this comment. The article type has been changed in the revised Version of the manuscript to „Short Communication “, according to the reviewer’s 1 suggestion
(Page 1)
Reviewer 2 Report
Comments and Suggestions for Authors
1. What are the future implications of this work, please discuss and justify.
2. Authors talking about inflammation in the abstract. How is this correlated with the inflammation? Please discuss and justify.
3. What inflammation markers did authors use to correlate? Also, there is no analysis in the manuscript that shows a relation with inflammation.
4. Haemophilus parainfluenzae is very well associated with upper respiratory infections and inflammation as well. Please discuss this by describing a little about the gut-lung axis. Here is one reference. https://pubmed.ncbi.nlm.nih.gov/33425362/
5. References are completely missing from material and methods. Please update.
6. Please make the limitation under a separate heading.
Comments on the Quality of English Language
This needs minor improvements.
Author Response
Reviewer 2
What are the future implications of this work, please discuss and justify
Response We thank Reviewer 2 for his/her criticism. The last part of discussion (Line 379-390, p10-11) as well the conclusion part (Line 415-421, p12) are dedicated exactly to this important issue. We do not know yet if anti-inflammatory effect of anthocyanins is mediated solely through mitigation of Haemophilus genus in IBD patients. Further (large-scale) relevant research is warranted before implementing this knowledge to clinical practice.
- Authors talking about inflammation in the abstract. How is this correlated with the inflammation? Please discuss and justify.
Response: In the “Conclusion” section of the abstract we indeed state “Our results suggest that ACRE treatment mitigates the contribution of H. parainfluenzae to inflammation.” Here, we are making reference to the measurements of fecal calprotectin. To make this clear, we have added the following to the abstract:
“We also correlated the relative abundances of individual microbial taxa at different time points to faecal concentrations of calprotectin, a proxy for inflammation.”
A relevant paragraph has been now inserted in the discussion part of the manuscript, according to Reviewer’s suggestion (Lines 331 - 346, p. 9)
- What inflammation markers did authors use to correlate? Also, there is no analysis in the manuscript that shows a relation with inflammation.
Response: Please kindly refer to previous comment (2.)
- Haemophilus parainfluenzae is very well associated with upper respiratory infections and inflammation as well. Please discuss this by describing a little about the gut-lung axis. Here is one reference. https://pubmed.ncbi.nlm.nih.gov/33425362/
Response: We appreciate this valuable comment. In the revised version of our manuscript, we have included a relevant paragraph (lung-gut axis) citing also the previously mentioned article, according to reviewer’s 2 recommendation:
Pathogenic bacteria of the upper respiratory tract like Haemophilus can interact with the gut microbiota along the so-called gut-lung axis. After birth, both the lungs and the gut are exposed to orally ingested microorganisms that contribute to the shaping of a stable and complex equilibrium between lung and gut flora. Gut microbes influence immune responses locally, systemically, and in the lungs, affecting conditions like asthma, allergic responses, and chronic obstructive pulmonary disease. Of note, in a relevant recent study with lung adenocarcinoma patients, Haemophilus parainfluenzae was the most commonly found species shared between the lung and gut microbiota.
(Line 339 – 346, p. 9)
- References are completely missing from material and methods. Please update.
Response: In the revised manuscript we have augmented the methods section with additional references where relevant.
(Lines 108 – 191, p 3 -5)
- Please make the limitation under a separate heading.
Response: Done, according to the reviewer´s 2 Suggestion
(Line 404, p. 11)
Reviewer 3 Report
Comments and Suggestions for Authors
Some suggestions:
1) In the introduction, I would add (lines 45-52) that, while it is true that biologics are associated with a non-negligible rate of adverse events (hence the valid point about the need for safer natural compounds), they still ensure significant levels of radical mucosal healing. I recommend citing a paradigmatic real-life example, such as ustekinumab in IBD, which can provide optimal rates of both mucosal healing and transmural healing, even in hard-to-treat patients (prior to colectomy):
Miranda, A.; Gravina, A.G.; Cuomo, A.; Mucherino, C.; Sgambato, D.; Facchiano, A.; Granata, L.; Priadko, K.; Pellegrino, R.; de Filippo, F.R.; et al. Efficacy of Ustekinumab in the Treatment of Patients with Crohn’s Disease with Failure to Previous Conventional or Biologic Therapy: A Prospective Observational Real-Life Study. J Physiol Pharmacol 2021, 72, 537–543, doi:10.26402/jpp.2021.4.05.
2) Lines 54-55: Be cautious about saying “herbal remedy” for UC; this seems somewhat overstated given the current level of evidence for anthocyanins. We are not talking about e.g. widely validated probiotics that help induce remission.
3) Introduce in the introduction that there are other natural substances emerging in both preclinical and clinical models, as in the case of medicinal mushrooms (I recommend discussing Hericium erinaceus as an example).
4) Reference is made to a Phase IIa RCT (in lines 76-80) as if it were already published, but it seems actually a pre-print. This should be clarified, as the work has not yet been validated by proper peer review and editorial acceptance.
5) Line 98: Specify the modified Mayo score with a reference. It is presumed to refer to the partial Mayo score, so include a reference for clarity and provide descriptors for the 6-12 scores (e.g., mild-moderate, moderate-severe activity).
6) Line 98-99: Were the patients taking steroids? Specify here what the maximum allowed dosage was for inclusion.
Author Response
Reviewer 3
1) In the introduction, I would add (lines 45-52) that, while it is true that biologics are associated with a non-negligible rate of adverse events (hence the valid point about the need for safer natural compounds), they still ensure significant levels of radical mucosal healing. I recommend citing a paradigmatic real-life example, such as ustekinumab in IBD, which can provide optimal rates of both mucosal healing and transmural healing, even in hard-to-treat patients (prior to colectomy):
Miranda, A.; Gravina, A.G.; Cuomo, A.; Mucherino, C.; Sgambato, D.; Facchiano, A.; Granata, L.; Priadko, K.; Pellegrino, R.; de Filippo, F.R.; et al. Efficacy of Ustekinumab in the Treatment of Patients with Crohn’s Disease with Failure to Previous Conventional or Biologic Therapy: A Prospective Observational Real-Life Study. J Physiol Pharmacol 2021, 72, 537–543, doi:10.26402/jpp.2021.4.05.
Response: We cannot agree more with Reviewer 3. Biologics are often very efficacious, the major problem is indeed the safety profile. We have inserted almost intact the truly well-expressed comment of Reviewer 3 in the revised manuscript and cited, appropriately.
(Lines 50 – 53, p. 2)
2) Lines 54-55: Be cautious about saying “herbal remedy” for UC; this seems somewhat overstated given the current level of evidence for anthocyanins. We are not talking about e.g. widely validated probiotics that help induce remission.
Response: We agree with this fair comment of Reviewer 3. In the revised version of the manuscript, we formulated more cautiously this sentence as following:
“Anthocyanins (AC) a specific type of deglycosylated anthocyanidins, are an alternative natural compound being investigated for potential UC treatment.”
(Lines 64 – 65, p.2)
3) Introduce in the introduction that there are other natural substances emerging in both preclinical and clinical models, as in the case of medicinal mushrooms (I recommend discussing Hericium erinaceus as an example).
Response: Done, according to reviewer’s 3 suggestion. The following text has been incorporated in the revised manuscript:
For instance, medicinal mushrooms have been demonstrated to exert beneficial actions against IBD. Specifically, Lion’s mane (Hericium erinaceus) is an edible fungi known to process anti-inflammatory as well as anti-neoplastic properties in colorectal tissue, which might be attributed to immune system regulation and gut microbiota. Particularly, this action might be mediated though a shift towards a phenotype with upregulation of short-chain fatty acids-producing bacteria.
(Lines 58-63, p.2)
4) Reference is made to a Phase IIa RCT (in lines 76-80) as if it were already published, but it seems actually a pre-print. This should be clarified, as the work has not yet been validated by proper peer review and editorial acceptance.
Response: We agree with the raised criticism by Reviewer 3. In the revised manuscript, we added the following Information to clarify that the RCT is still under Peer-Review process:
“In the phase IIa study, which is currently under Peer-Review process and as pre-Print available, we observed that faecal calprotectin concentrations decreased during the treatment phase and subsequently increased again after stopping ACRE therapy.”
(Lines 97-99, p. 3)
5) Line 98: Specify the modified Mayo score with a reference. It is presumed to refer to the partial Mayo score, so include a reference for clarity and provide descriptors for the 6-12 scores (e.g., mild-moderate, moderate-severe activity).
Response: Done according to the reviewer’s suggestion. The following paragraph has been inserted in the methods section for better clarification:
Of note, the modified Mayo score has been initially introduced by food and drug administration as “Guidance for Clinical Trial Endpoints”. Specifically, it refers to the „Endoscopy subscore” of the Mayo Score, which should be modified so that a value of 1 does not include friability. This is due to the fact that existence of friability (even if graded to be mild by the endoscopist/central reader) is not consistent with clinical remission). A Mayo score of 5 or below indicates mild disease activity, a score between 6 and 10 signifies moderate activity, and a score from 11 to 12 represents severe disease activity.
(Lines 120 – 127, p. 3)
6) Line 98-99: Were the patients taking steroids? Specify here what the maximum allowed dosage was for inclusion.
Response: Thank you for asking this meaningful question. We have already addressed this important issue. The information can be found in the Supplementary Table of Appendix under “inclusion criteria”. For your convenience we also attach the relevant text below. Upon your request, it can also be transferred to main body of text.
Allowed to receive a therapeutic dose of following UC drugs during the study:
- Oral steroids therapy (≤30mg prednisone or equivalent/day) providing that the dose has been stable for 2 weeks prior to baseline
- Oral or rectal MMX Budesonide therapy (9mg/day) initiated at least 8 weeks before baseline
- Oral or rectal 5-ASA/SP compounds, providing that the dose has been stable for 2 weeks before baseline visit
- AZA/6-MP providing that the dose has been stable for 8 weeks prior to baseline and been initiated at least 2 months before screening
(Page 13)
Reviewer 4 Report
Comments and Suggestions for Authors
Dear Editor, Dear Authors,
I was invited to evaluate the following manuscript :
« Anthocyanin-rich extract mitigates the contribution of the pathobiont genus Haemophilus in mild-to-moderate ulcerative colitis patients » by Yannik Zobrist et al.
In this study, the authors investigated the effects of anthocyanins (AC) on ulcerative colitis (UC). AC are known as anti-inflammatory and antioxidant molecules and to improve ulcerative colitis (UC) in animal models and humans. Here, the authors investigated and reported on the changes in the fecal microbiome composition in the patient upon AC exposure. For that, the authors gave to UC patients a 3g daily dose of an AC-rich bilberry extract (ACRE) for eight weeks and determined the microbiome composition in longitudinal stool samples from 24 patients for quantification of the degree of change over time with quantification of the relative abundances of individual microbial taxa at different time points to fecal concentration measurements of calprotectin. The data provided show that the microbiome compositions did not change over time of treatment. However, contrarly to control and placebo, the correlation between H. parainfluenzae and the concentration of calprotectin vanished in ACRE-treated subjects, interestingly without alteration of the relative abundance of H. parainfluenzae. The authors to conclude that their results suggest that ACRE treatment mitigates the contribution of H. parainfluenzae to inflammation as demonstrated by calprotectin marker suggesting that AC-rich extract should be considered to treat UC patients.
I found the data strong and convincing, with conclusions mostly supported by the data.
I have only few comments :
1- I will suggest to further explain in introduction and more importantly in conclusion if and how AC acts as antiinflammatory molecules. Indeed, the authors concluded that the effect of AC in UC is due to the fact that AC increase IgA production. In my opinion, IgA increase is a sign of IBD (many IgA are increased in IBD including UC), so it is hard to understand how AC, by causing further IgA secretion, may protect against UC in which IgA secretion is already high.
2- Does the antioxidant effect of AC may play a role in decrease in calprotectin ? Either directly or indirectly by decreasing inflammation as observed with many antioxidants ?
3- The data suggest no effect on microbiota diversity neither on H parainfluenzae count. What about other important bacteria associated to IBD such as MAP or AIEC ?
4- Do the authors have the opportunity to measure also inflammation markers in the blood of the patients ? And also bacterial count in their blood as IBD is associated to increase in bacterial count in blood of patient.
5- Although fecal Calprotectin is a known and well admitted marker of IBD progression, could the authors also measure other fecal parameters and blood parameters of inflammation ?
regards
Author Response
Reviewer 4
I was invited to evaluate the following manuscript :
« Anthocyanin-rich extract mitigates the contribution of the pathobiont genus Haemophilus in mild-to-moderate ulcerative colitis patients » by Yannik Zobrist et al.
In this study, the authors investigated the effects of anthocyanins (AC) on ulcerative colitis (UC). AC are known as anti-inflammatory and antioxidant molecules and to improve ulcerative colitis (UC) in animal models and humans. Here, the authors investigated and reported on the changes in the fecal microbiome composition in the patient upon AC exposure. For that, the authors gave to UC patients a 3g daily dose of an AC-rich bilberry extract (ACRE) for eight weeks and determined the microbiome composition in longitudinal stool samples from 24 patients for quantification of the degree of change over time with quantification of the relative abundances of individual microbial taxa at different time points to fecal concentration measurements of calprotectin. The data provided show that the microbiome compositions did not change over time of treatment. However, contrarly to control and placebo, the correlation between H. parainfluenzae and the concentration of calprotectin vanished in ACRE-treated subjects, interestingly without alteration of the relative abundance of H. parainfluenzae. The authors to conclude that their results suggest that ACRE treatment mitigates the contribution of H. parainfluenzae to inflammation as demonstrated by calprotectin marker suggesting that AC-rich extract should be considered to treat UC patients.
I found the data strong and convincing, with conclusions mostly supported by the data.
We sincerely thank Reviewer 4 for the encouraging and constructive comments
I have only few comments:
1- I will suggest to further explain in introduction and more importantly in conclusion if and how AC acts as antiinflammatory molecules. Indeed, the authors concluded that the effect of AC in UC is due to the fact that AC increase IgA production. In my opinion, IgA increase is a sign of IBD (many IgA are increased in IBD including UC), so it is hard to understand how AC, by causing further IgA secretion, may protect against UC in which IgA secretion is already high.
Response: We thank reviewer 4 for the excellent observation. We agree that secretory IgA has a crucial pivotal role in homeostasis and IBD although the exact pathogenetical role is not fully elucidated, yet. We added relevant sentences to introduction and conclusion parts for better clarification.
Introduction addition: (Lines 83-92, p. 2-3): A further possible key-attribute of AC as antioxidant and anti-inflammatoryagent on IBD might be mediated via secretory immunoglobulin A (IgA). IgA is produced by antibody secretory cells locally in gut lumen and forms dimers. IgA possesses a crucial pivotal role in shaping the gut microbiome composition and maintain homeostasis with the intestinal immune system. On the other hand, derangements in IgA production, secretion, and/or function may occur during pathological conditions such as IBD, the pathogenesis of which remains largely not-elucidated. In a relevant clinical study, oral ACRE administration was beneficial for the management of oxidative stress and inflammation, and effect that was attributed to increase of IgA, antimicrobial beta-defensin 2 as well as anti-inflammatory IL-10.
Conclusion addition: (Lines 415 – 419, p. 11) Given IgA’s dual role in maintaining gut homeostasis and its involvement in inflammatory processes in IBD, it is plausible that ACs exert anti-inflammatory effects through the modulation of IgA secretion. This action could support intestinal immune balance, potentially attenuating inflammation. Nonetheless, the specific interactions between AC intake, IgA dynamics, and the microbiome in UC are not yet fully elucidated, underscoring the need for further research to clarify these mechanisms
2- Does the antioxidant effect of AC may play a role in decrease in calprotectin ? Either directly or indirectly by decreasing inflammation as observed with many antioxidants ?
Response: Thank you for this truly excellent mechanistical question about AC role on calprotectin (i.e. inflammation). The honest direct answer is that we do not know exactly, it is however reasonable to postulate that this action is mediated through downregulation of proinflammatory cytokines such as interleukins 1b and 6, tumor necrosis alpha and upregulation of anti-inflammatory interleukin 10. We have now inserted a relevant paragraph in discussion section, just after the hypothesis of how AC act on gut microbiome, directly or indirectly.
(Lines 221-225, p. 6)
3- The data suggest no effect on microbiota diversity neither on H parainfluenzae count. What about other important bacteria associated to IBD such as MAP or AIEC ?
Response: Please refer to Figure 2 as well as clinical study protocol (attached) for the genera that have been investigated within this work. Due to financial (grant) and ethical (ethical approval) restrictions, a further experiment after the completion of the study, is unfortunately no more possible.
(Line 261, p. 7)
4- Do the authors have the opportunity to measure also inflammation markers in the blood of the patients ? And also bacterial count in their blood as IBD is associated to increase in bacterial count in blood of patient.
Response: Please kindly refer to last comment of yours as well second comment of second reviewer. Regarding the bacterial count, we did not measure any bacteria in the bloodstream of the patients, since blood is normally sterile and free of any microorganisms and a circulation of bacteria along with systemic inflammation i.e. also with accompanying fever is a sepsis/septicaemia setting, which was outside the scope of our research
5- Although fecal Calprotectin is a known and well admitted marker of IBD progression, could the authors also measure other fecal parameters and blood parameters of inflammation ?
Response: As stressed already before (please kindly refer also to the replies of rest reviewers’ as well), we did measure the most validated inflammation markers in blood (CRP) and stool (faecal calprotectin). Given the fact that this RCT had to be prematurely terminated due to COVID-19 pandemics as well as no more grant to support further investigations was available, such measures several years of completion of the RCT is unfortunately not possible.